# Chinese Admission Warning Strategy for Predicting the Hospital Discharge Outcome in Patients with Traumatic Brain Injury

**DOI:** 10.3390/jcm11040974

**Published:** 2022-02-13

**Authors:** Ruizhe Zheng, Zhongwei Zhuang, Changyi Zhao, Zhijie Zhao, Xitao Yang, Yue Zhou, Shuming Pan, Kui Chen, Keqin Li, Qiong Huang, Yang Wang, Yanbin Ma

**Affiliations:** 1Hospital Development Center, Shanghai 200041, China; Ruizhedoctor@126.com; 2Department of Neurosurgery, Huashan Hospital, Fudan University, Shanghai 200040, China; 3Department of Neurosurgery, Shanghai East Hospital, Tongji University School of Medicine, Shanghai 200120, China; 13917920669@163.com (Z.Z.); dr_chen2017@163.com (K.C.); ldszz@163.com (K.L.); 4Department of Neurosurgery, Tongren Hospital, Shanghai Jiao Tong University School of Medicine, Shanghai 200336, China; zcy3712@shtrhospital.com (C.Z.); 15836015662@163.com (Z.Z.); 5Department of Interventional Therapy, Shanghai Ninth People’s Hospital, Shanghai Jiao Tong University School of Medicine, Shanghai 201999, China; xitao123456@126.com; 6Department of Automation, School of Electronic Information and Electrical Engineering, Shanghai Jiao Tong University, Shanghai 200240, China; zhouyue@sjtu.edu.cn; 7Department of Emergency, Xinhua Hospital, Shanghai Jiao Tong University School of Medicine, Shanghai 200092, China; panshuming@xinhuamed.com; 8Department of Neurology, Tongren Hospital, Shanghai Jiao Tong University School of Medicine, Shanghai 200336, China; hq2716@shtrhospital.com

**Keywords:** warning strategy, admission, hospital discharge outcome, traumatic brain injury, emergency

## Abstract

Objective: To develop and validate an admission warning strategy that incorporates the general emergency department indicators for predicting the hospital discharge outcome of patients with traumatic brain injury (TBI) in China. Methods: This admission warning strategy was developed in a primary cohort that consisted of 605 patients with TBI who were admitted within 6 h of injury. The least absolute shrinkage and selection operator and multivariable logistic regression analysis were used to develop the early warning strategy of selected indicators. Two sub-cohorts consisting of 180 and 107 patients with TBI were used for the external validation. Results: Indicators of the strategy included three categories: baseline characteristics, imaging and laboratory indicators. This strategy displayed good calibration and good discrimination. A high C-index was reached in the internal validation. The multicenter external validation cohort still showed good discrimination C-indices. Decision curve analysis (DCA) showed the actual needs of this strategy when the possibility threshold was 0.01 for the primary cohort, and at thresholds of 0.02–0.83 and 0.01–0.88 for the two sub-cohorts, respectively. In addition, this strategy exhibited a significant prognostic capacity compared to the traditional single predictors, and this optimization was also observed in two external validation cohorts. Conclusions: We developed and validated an admission warning strategy that can be quickly deployed in the emergency department. This strategy can be used as an ideal tool for predicting hospital discharge outcomes and providing objective evidence for early informed consent of the hospital discharge outcome to the family members of TBI patients.

## 1. Introduction

Affecting approximately 18% of the world population and with an annual incidence of over 50 million, the absolute numbers of patients with traumatic brain injury (TBI) in China have exceeded those of most other countries [1,2]. Recently, a greater number of resources and advanced facilities have become available for the diagnosis and treatment of TBI. Consequently, the overall mortality rates of severe TBI patients in China have gradually decreased and are similar to those reported in most Western countries [2]. However, with an approximately 27% mortality and with over 50% having an unfavorable outcome of severe TBI, TBI is still a major concern in China [2,3,4].

With improvements in TBI management in China (e.g., the issuing of a series of guidelines and consensus reports) [1], the treatment level of TBI in the majority of hospitals is normalizing, and increasing numbers of senior neurologists have the ability to predict the outcome of TBI. Moreover, fast prognosis prediction helps to guide long-term planning, assess the effectiveness of clinical management, and reasonably regulate and assign medical resources [5]. Thus, the early prognostic assessment of TBI is deeply embedded in routine clinical practice [6]. Over the years, some predictors, such as patient demographics, imaging findings, clinical presentation, and fluid biomarkers, have been proven to have predictive value [7,8,9,10]. Currently, there is increasing interest in the discovery of novel biomarkers and the development of advanced technical means for predicting outcomes following TBI [11,12,13,14]. These indicators may accelerate the development of more accurate prediction strategies that could be useful for determining patient outcomes [7,15,16]. However, considering their time-consuming and exhausting nature, the execution of sophisticated or repetitive detection technologies is impractical in the emergency setting. The lack of precise information and heterogeneity in study design and statistical analyses present additional challenges [6].

In China, from the initial admission to hospitalization in a specialized department, procedures for emergency triage are often required. During this period, neurologists are often pressed to prognosticate patient outcomes, such as mortality or other unfavorable outcomes during an emergency consultation. To our knowledge, the most widely used traditional prognostic predictors at this stage are the injury details, clinical severity, and GCS score [17,18,19]. Meanwhile, several computed tomographic (CT) severity grading scales (e.g., Marshall CT score, etc.) were used to develop models for TBI outcome prediction [20]. Unfortunately, the accuracy of these predictors or scales does not meet the need of providing clinicians with an early warning so they can provide informed consent for hospital discharge to a patient’s family. Gradually, clinicians have recognized that a multidimensional approach can achieve clinically relevant accuracy for providing prognostic information in TBI patients [21]. Landmark analyses in a recent study found that injury severity characteristics and physiological monitoring may emerge as prognostic predictors during the first day post-injury [22]. However, the authors failed to provide effective information about the emergency department studied, which is necessary given that neurologists are expected to give an early warning signal before hospitalization.

Therefore, an efficient solution to reintegrate examination indicators is not only necessary but also urgent against this backdrop. A possible path is to employ a reasonable algorithm and then design a multidimensional strategy [23]. The purpose of this study was to develop and validate an admission warning strategy in the emergency department by incorporating admission baseline characteristics and initial examination indicators to predict the hospital discharge outcome of TBI patients.

## 2. Patients and Methods

### 2.1. Participants

This study enrolled patients with TBI who were transported to the emergency department by ambulance in three subcenters (Shanghai Tongren Hospital, Shanghai East Hospital, and Xinhua Hospital). These subcenters were randomly divided into three independent cohorts: one training and internal validation cohort, and two external validation cohorts. The inclusion criteria for the three cohorts were: TBI patients with no previous history of brain injury, admission within 6 h of injury, age ≥ 18 years. The exclusion criteria were: death before the admission examination, pregnancy, or the coexistence of other fatal diseases. This multicenter study was approved by the ethics committee of all participating institutions and the need to obtain informed consent was waived for the individual patients. The research was carried out before the outbreak of COVID-19.

### 2.2. Data Collection and Patient Management

We collected routine examination indicators in the emergency departments of three subcenters in China [6,7,15]. Each TBI patient’s baseline characteristics, including age, sex, time from injury to admission, mechanism of injury, injury type, and admission GCS score, as well as other characteristics, including the patient’s motor score, pupil condition, vital signs, basic diseases, and complications were recorded, and non-contrast CT imaging and laboratory indicators were obtained.

All patients with TBI were treated according to the current institutional strategy based on the Brain Trauma Foundation Guidelines, 4th edition [24] during hospitalization. All patients were managed by a team of neurosurgeons with the title of senior professional post and full-time practitioner. TBI comorbidities, such as extracranial injury, underlying diseases, hypoxemia, hypotension, pneumonia, and electrolyte disturbance were treated with individualized and precise strategies in all TBI patients.

### 2.3. Outcome Assessment

Each patient’s outcome was assessed on the date when the patient met discharge criteria [24]. We used the Extended Glasgow Outcome Scale (GOSE) scale (where 1 = died and 8 = totally recovered), which increased the original five categories of the traditional GOS to eight categories by dividing the latter three categories into upper and lower bands, as follows: dead, vegetative state, lower/upper severe disability, lower/upper moderate disability, and lower/upper good recovery [17]. The TBI patient outcomes were dichotomized into “favorable outcome”, defined as a GOSE score of 5–8 points, and “unfavorable outcome”, defined as a GOSE score of 1–4 points.

### 2.4. Statistical Analysis (Developing and Validating Phase)

Statistical computing and graphics were performed with R software (https://www.r-project.org, Shanghai Jiao Tong University, Shanghai China, v3.5.0; accessed on 1 December 2021) and SPSS software (China, Chinese, v22.0) with *p* value less than 0.05 considered as statistical significance. (1) In the developing phase, the least absolute shrinkage and selection operator (LASSO) was used for data dimension reduction and the selection of optimal predictive features from initial admission indicators [25]. Next, multivariable logistic regression analysis was used to develop an early warning strategy by incorporating the features selected in the LASSO regression mode. The significant results are reported as two-sided odds ratios (ORs) with 95% confidence intervals (CIs) and as *p*-values < 0.05. The continuous variables are expressed as the means ± standard deviations (means ± SDs) or medians and interquartile ranges (IQRs). All routine indicators available in the China emergency department were collected during the strategy development phase [26]. (2) In the validation phase, we plotted calibration curves to determine the difference between predicted value and actual value, and measured Harrell’s C-index to evaluate the performance of our strategy (developed in primary cohort). Bootstrapping validation (1000 bootstrap resamples) was used for internal validation and to calculate the C-index [27]. Decision curve analysis (DCA) was performed to determine the clinical usefulness of our strategy. In DCA, the horizontal line indicates that all samples are negative with a net benefit of 0, and the slash indicates that all samples are positive with a net benefit of 1 [28]. MedCalc software (New York, NY, USA, v20.0.22) was used to perform receiver operating characteristic (ROC) analysis and calculate the area under the ROC curve (AUC). The optimal thresholds for the parameters used to predict patient outcomes were defined as the optimal cutoff points on the ROC curve when the maximal Youden’s index was obtained. High accuracy was defined as C-index or AUC > 0.9. Finally, pairwise comparison analysis was used for evaluating the difference in performance between our strategy and traditional indicators. The same process was carried out in the three cohorts.

## 3. Results

### 3.1. Participants

The patients’ demographic and admission clinical characteristics, outcomes, and routine examination indicators at admission are summarized in Table 1 (for more detail, see Appendix A). In total, 892 patients with TBI were assessed for eligibility between January 2018 and May 2019, for whom no significant difference was observed according to the clinical characteristics between the internal validation cohort (605 patients) and the two external validation cohorts (180 and 107 patients).

### 3.2. Feature Selection and Admission Warning Strategy Development

Focusing on the selected admission indicators, a total of eighteen emergency indicators with non-zero coefficients were selected in the LASSO logistic regression analysis. After the application of the LASSO logistic algorithm, 10 out of the 18 indicators were eventually used to develop the strategy (Figure 1). They were the baseline characteristics (age, GCS score, pupillary reactivity and hypotension), imaging indicators (midline shift, intracerebral hematoma, traumatic subarachnoid hematoma, and basal cistern), and laboratory indicators (glucose level and monocyte count).

Subsequently, multivariable logistic regression analysis was used to evaluate the LASSO selected features (Table 2). When *p* < 0.05, LASSO selected indicators can be used as the predictors of unfavorable outcomes.

Next, a nomogram model was formulated based on the ability of these predictors. As shown in Figure 2, this warning strategy can be posted in the general emergency department.

## 4. Clinical Use and Validation

### 4.1. Internal Validation

The C-index of our strategy was 0.982 (95% CI: 0.973–0.990) and 0.975 (95% CI: 0.966–0.983) according to the bootstrapping analysis, suggesting that the strategy had good discriminative ability. The calibration plots of the strategy showed that the agreement between predicted and observed unfavorable outcomes at hospital discharge of patients after TBI was optimal (Figure 3A). The DCA showed that the strategy had significant net benefits for all the threshold probabilities for admission indicators, and the potential clinical benefits of the prediction model were demonstrated (Figure 3B). The AUC of the warning model predicting an unfavorable outcome at the hospital discharge was 0.981 (95% CI: 0.967–0.991), indicating improved survival prediction compared with the traditional predictive model (Figure 3C). The multivariate logistic regression analysis showed that the following variables were predictors of unfavorable outcomes (*p <* 0.05). The performance of these independent predictors was: age (AUC = 0.633; 95% CI: 0.593–0.671), GCS score (AUC = 0.922; 95% CI: 0.898–0.942), Marshall CT score (AUC = 0.854; 95% CI: 0.824–0.881), glucose (AUC = 0.786; 95% CI: 0.751–0.818), and monocyte count (AUC = 0.688; 95% CI: 0.649–0.724) (Figure 3D–H). Finally, the pairwise comparison of the ROC curves farther indicated that there was a statistically significant relationship (*p* < 0.05) between the strategy and the age, GCS score, Marshall CT score, glucose, and monocyte count.

### 4.2. External Validation

Multicenter patients with TBI (180 patients in sub-cohort one and 107 patients in sub-cohort two) were used for external validation. The calibration curves for our strategy showed that the external use of predicting unfavorable outcomes at hospital discharge of patients after TBI is always appropriate (Figure 4A,B). The C-index of the strategy was 0.926 (95% CI: 0.872–0.979) in sub-cohort one and 0.959 (95% CI: 0.923–0.994) in sub-cohort two. DCA showed that the strategy was clinically useful when the possibility threshold was 0.02–0.83 for sub-cohort one and 0.01–0.88 for sub-cohort two. The predictive capacity of the warning model was shown as ROC curves for the prediction of TBI outcome, and the AUCs of the two sub-cohorts were 0.926 (95% CI: 0.877–0.959) and 0.959 (95% CI: 0.902~0.988) (Figure 4E,F), respectively. In addition, this strategy exhibited a better prognostic performance than the independent predictor: age (AUC = 0.514; 95% CI: 0.439~0.589), GCS score (AUC = 0.877; 95% CI: 0.820–0.921), Marshall CT score (AUC = 0.760; 95% CI: 0.691–0.820), glucose (AUC = 0.729; 95% CI:0.658–0.793), and monocyte count (AUC = 0.611; 95% CI:0.535–0.682) for sub-cohort one and age (AUC = 0.536; 95% CI: 0.437–0.633), GCS score (AUC = 0.894; 95% CI: 0.820–0.945), Marshall CT score (AUC = 0.750; 95% CI:0.657–0.829), glucose (AUC = 0.732; 95% CI: 0.638–0.813), and monocyte count (AUC = 0.616; 95% CI: 0.517–0.708) for sub-cohort two (Figure 4E,F). The pairwise comparison of the ROC curves further indicates that there was statistically significant relationship (*p* < 0.05) between the strategy and the age, GCS score, Marshall CT score, glucose, and monocyte count in the two sub-cohorts.

## 5. Discussion

Although several basic multiple trauma scores, such as the revised trauma score (RTS), injury severity score (ISS), exponential injury severity score (EISS), and traumatic injury mortality prediction (TRIMP), have been widely accepted, they are so influenced by the anatomical index and the accuracy of specific anatomical injury site (brain injury) is limited [29]. Gradually, there have been many prognostic models incorporating the different parameters to predict TBI patients’ outcomes, producing a predictive value range from acceptable to good [7,15,16]. Taking a traditional TBI-based model as an example, incremental increases in predictive value were achieved, with AUCs ranging from 0.74 for the core model (basic characteristics) to 0.77 with the CT indicators added and 0.79 with the laboratory indicators added for the TBI patient outcomes. As the treatment of TBI tends to be standardized at each tertiary hospital in China [1,2,24], it is increasingly important to show the importance of using initial indicators in TBI prediction. Thus, our hypothesis was that a multidimensional modeling strategy utilizing patients’ initial admission indicators would improve outcome prediction compared with a traditional independent predictor.

Hence, we collected TBI patient admission indicators when the indicators met the following conditions: (1) they can be quickly obtained, (2) they are essential for diagnosis, and (3) they are available in the majority of emergency departments in China. Therefore, the data of 605 patients in emergency departments were suitable for developing a warning strategy, and age, GCS score, blood pressure, pupillary reactivity, midline shift, intraparenchymal lesion, traumatic subarachnoid hemorrhage, basilar cistern, blood glucose levels, and monocyte count were adopted to obtain unified dimensions by shrinking the regression coefficients with the LASSO method. We chose the predictors based on the weight of their univariable association with TBI outcomes and combined those selected features into our admission warning strategy. We found that the elderly population suffered dramatically worse long-term outcomes than the younger population despite less-severe initial head injuries. The selected factor for increased unfavorable outcome risk in our study was age > 67 years and hypotension, defined as a systolic blood pressure (SBP) < 90 mmHg. These results are in line with previous single-center studies that we reviewed [30,31]. Additionally, other admission baseline characteristics, such as GCS score and pupillary reactivity, are traditionally considered unfavorable indicators. In terms of supplementary imaging indicators, we found that intracerebral hemorrhage (ICH), traumatic subarachnoid hematoma (tSAH) and basal cistern (obliteration or compressed) were incremental indicators that improved outcome prediction accuracy [32,33,34,35]. In addition, adding invasive laboratory indicators, such as blood glucose and circulating monocyte count, will substantially improve the accuracy [36,37,38].

Traditionally, the best contributor to the prediction of unfavorable TBI-related outcomes was the GCS score at admission [22]. Previous researchers found that the GCS score post-baseline was an important predictor of TBI outcomes with an AUC of 0.960, and changes in GCS scores on days 7 and 14 postinjury were the most influential [39]. When this notion was applied to our strategy, the best GCS score had an AUC of 0.922, indicating good precision. However, a better AUC of 0.981 was obtained when combining the LASSO-selected multidimensional parameters into our admission warning strategy, which means that our strategy is better than the traditional prediction method. Moreover, compared with traditional prognostic factors, this warning strategy displayed significantly better discrimination than single-dimensional predictors, such as age, serum glucose level, and circulating monocyte count. In addition, this performance is unbiased because external validation also showed similar optimization. Considering that sedation and treatment-related conditions may affect the patient’s level of consciousness, changes in the GCS score during admission were not included in the study. Therefore, this study provides an ideal approach to the development of a multidimensional prediction strategy.

To the best of our knowledge, the traditional Marshall CT model was shown to contribute to outcome prediction in patients with TBI in previous studies [7,40]. In our study, we tested the validity of the Marshall CT model, and our strategy displayed significantly better discrimination than the Marshall CT model (*p* < 0.05). This means that the multidimensional strategy is more optimized than the traditional model. These results were in line with previous studies, where a more efficient three-dimensional scoring prediction model was proposed to predict the outcome of TBI patients with a high AUC value [41].

Although studies have shown that dynamic monitoring imaging parameters can significantly improve the prognostic value, it was convenient to integrate laboratory examination information into an independent strategy for most general emergency department in China. In this way, using emergency indicators alone has no added value and can be fully avoided, regardless of the influence of sample size [22]. Moreover, the heterogeneity of statistical methods, strategic differences in addressing missing data, and differences in the first derivatives of indicators account for the biased and unrealistic findings. Additionally, inconsistencies in modeling approaches and the lack of external multicenter validation have become common topics [6,22,42]. To address these problems, we selected the most general indicators according to the medical record system. No additional behaviors other than a routine examination were required, and two separate cohorts of patients were used for the external validation. The performance of our strategy was ideal and provided improved accuracy for the development of prediction models. In particular, this strategy fills a gap in the investigation of initial emergency examinations.

## 6. Limitations and Future Implications

The methods used in this study were aimed at developing and validating an admission warning strategy for predicting TBI patient outcomes. Nevertheless, some limitations of this study should be noted. First, only short-term outcomes (from emergency to hospital discharge) were included, and we failed to perform additional validation with regard to later follow-up outcomes. Second, the endpoint of our strategy was closer than that of traditional models, such as “CRASH” or “IMPACT”, and its superiority over these competing prediction models remains unclear. However, it can be simply promoted in China’s emergency department through process posters (while “CRASH” or “IMPACT” models require proprietary website settings). Thirdly, only a small number of medical institutions were included in the development of this strategy. The parameters in this model should be adjusted according to the different characteristics of different centers in the future. Moreover, this strategy does not include the pediatric TBI group, and there might be some performance degradation in our strategy after age adjustment. Finally, longer follow-up is still necessary because some important predictors may emerge in the process.

## 7. Conclusions

To our knowledge, this is an innovative proposal of initial admission indicators of TBI patients based on a relatively large sample size for strategy development and two independent samples for external evaluation in China. It would be worth exploring the screening and optimization of the emergency indicator system. This strategy can be used to provide objective evidence for clinical strategy development and the provision of accurate informed consent for hospital discharge to patients’ family members.

## Figures and Tables

**Figure 1 jcm-11-00974-f001:**
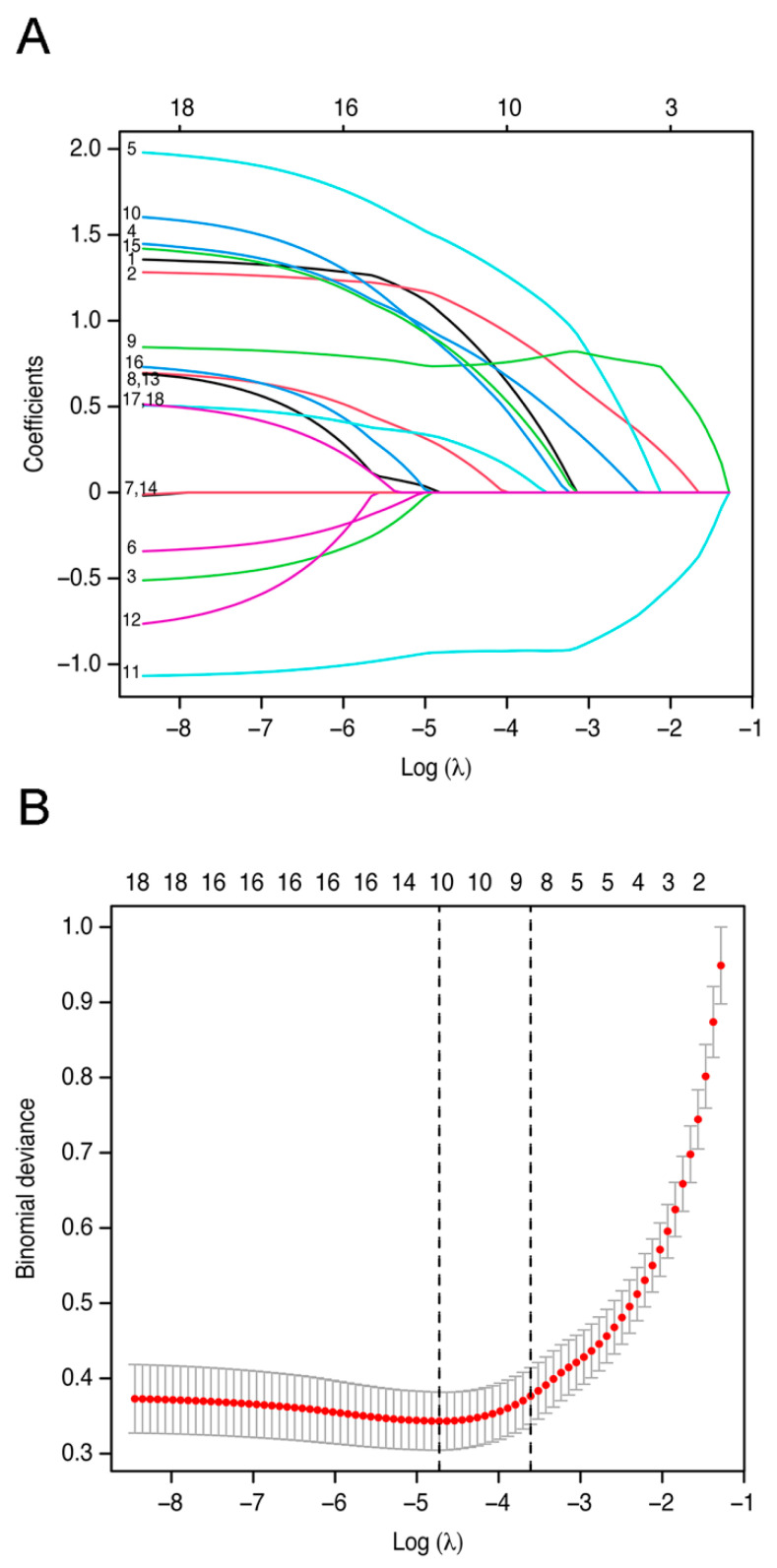
Indicators included in the model were selected using the least absolute shrinkage and selection operator (LASSO) binary logistic regression model. Figure legends: (**A**) LASSO coefficient profiles, displaying eighteen texture features. A coefficient profile plot was produced against the log (lambda) sequence. (**B**) Optimal parameter (lambda) selection in the LASSO model used fivefold cross-validation and minimum criteria. The partial likelihood deviance (binomial deviance) curve was plotted versus log (lambda). Dotted vertical lines were drawn at the optimal values by using the minimum criteria and the 1 standard error (SE) of the minimum criteria (the 1-SE criteria).

**Figure 2 jcm-11-00974-f002:**
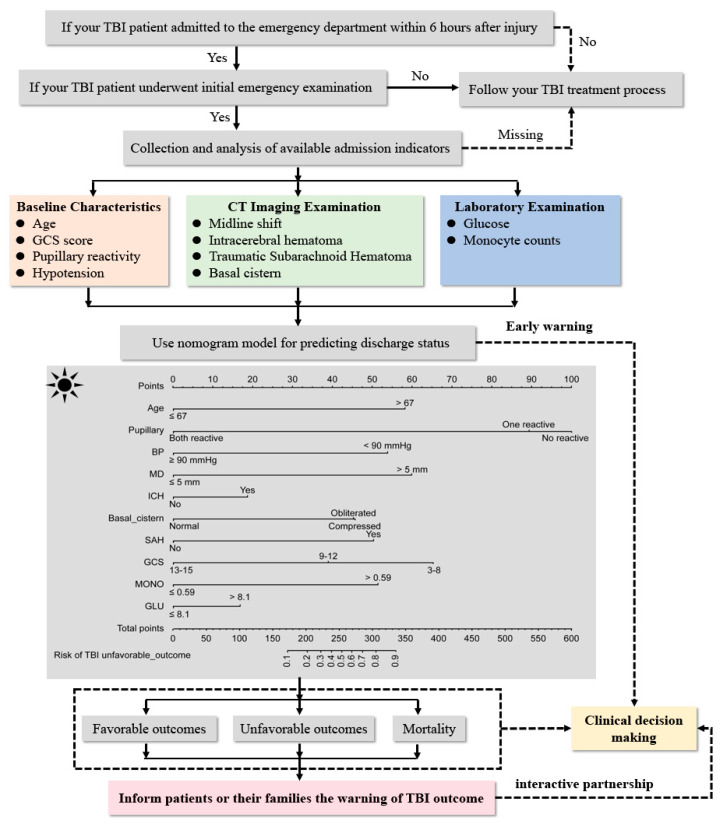
Description of admission warning strategy. Figure legends. An admission warning strategy incorporated the admission baseline characteristics and routine examination indicators, and the nomogram was developed in the primary cohort with the use of the independent predictors (TBI: traumatic brain injury; CT: computed tomography; GCS: Glasgow Coma Scale; BP: blood pressure; MD: midline shift; ICH: intracerebral hemorrhage; SAH: subarachnoid hematoma; MONO: monocyte count; GLU: glucose).

**Figure 3 jcm-11-00974-f003:**
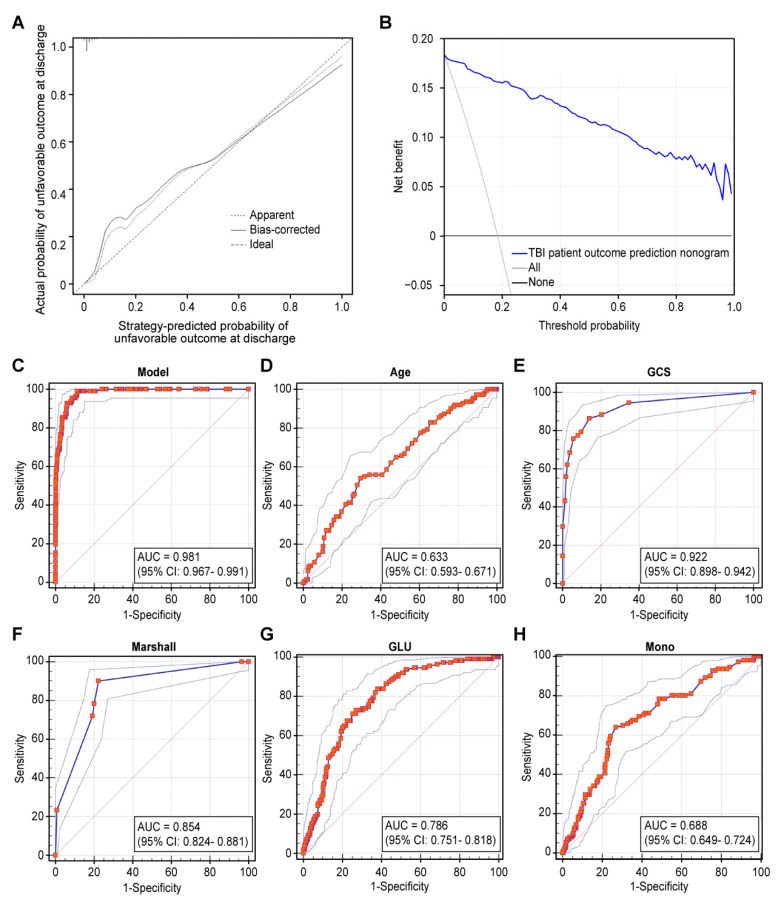
Internal validation of the warning strategy. Figure legends. (**A**) Calibration curves of the nomogram for predicting unfavorable outcomes for patients after TBI. Data on predicted unfavorable outcomes are plotted on the x- and y- axes. The diagonal dotted line indicates the ideal nomogram, in which actual and predicted probabilities are identical. The solid line represents the actual nomogram, and the higher the fitting degree with the dotted line was, the better the calibration effect would be. (**B**) Decision curves of the strategy predicting an unfavorable outcome at the threshed of 0.01. The x-axis represents the threshold probability, and the net benefit of the y-axis measurement was calculated by adding the true positive minus the false positive. (**C**) ROC of the warning strategy for predicting unfavorable outcomes. (**D**–**H**) ROC of the independent predictors for predicting unfavorable outcomes.

**Figure 4 jcm-11-00974-f004:**
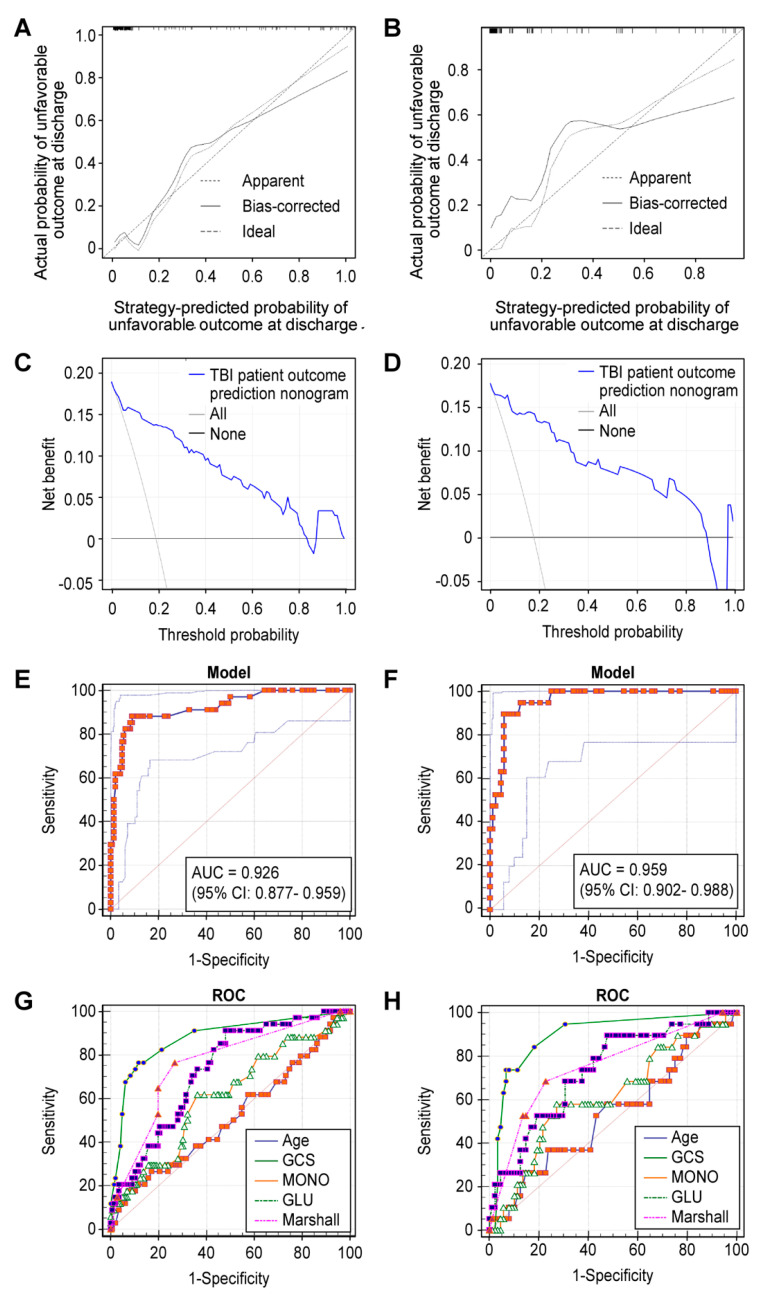
External validation of the warning strategy. Figure legends. Calibration curves for external validation in sub-cohort one (**A**) and sub-cohort two (**B**). Decision curves of the strategy predicting an unfavorable outcome in sub-cohort one (**C**) and sub-cohort two (**D**). ROC curves for the prediction of strategy in sub-cohort one (**E**) and sub-cohort two (**F**). ROC curve for the prediction of traditional predictors in sub-cohort one (**G**) and sub-cohort two (**H**).

**Table 1 jcm-11-00974-t001:** Patient demographics and admission clinical characteristics.

Variables	Primary Cohort (n = 605)	Sub Cohort One (n = 180)	Sub Cohort Two (n = 107)
Age (years) (mean ± sd)	60.1 ± 18.0	60.6 ± 17.0	59.3 ± 15.8
Sex (n, %)			
Male	401 (66.3%)	117 (65.0%)	61 (57.0%)
Female	204 (33.7%)	63 (35.0%)	46 (43.0%)
Mechanism of head injury (n, %)			
Traffic incident	242 (40.0%)	64 (25.6%)	
Fall	318 (52.6%)	96 (53.3%)	39 (36.4%)
Other cause	45 (7.4%)	20 (11.1%)	57 (53.3%)
Time from injury to admission (h) (median, iqr)	6 (3–12)	6 (3–12)	6 (3–12)
Pupillary reactivity at admission (n, %)			
Normal	536 (88.6%)	157 (87.2%)	92 (86.0%)
Unilateral abnormality	21 (3.5%)	10 (5.6%)	5 (4.7%)
Bilateral abnormality	48 (7.9%)	13 (7.2%)	10 (9.3%)
Gcs score at admission			
14–15	406 (67.1%)	121 (67.2%)	75 (70.1%)
9–13	104 (17.2%)	27 (15.0%)	15 (14.0%)
≤8	95 (15.7%)	32 (17.8%)	17 (15.9%)
Hypotension at admission (<90 mmhg) (n, %)			
Yes	61 (10.1%)	21 (11.7%)	9 (8.4%)
No	544 (89.9%)	159 (88.3%)	98 (91.6%)
Combined extracranial injuries (number) (mean ± sd)	1.3 ± 1.6	1.4 ± 1.6	1.2 ± 1.5
Combined underlying diseases (number) (mean ± sd)	0.9 ± 1.1	1.0 ± 1.2	0.9 ± 1.0
Neurosurgical procedure (n, %)			
Yes	145 (24.0%)	52 (28.9%)	20 (18.7%)
No	460 (76.0%)	128 (71.1%)	87 (81.3%)
GOSE at discharge (n, %)			
Favorable outcome for 5–8	494 (81.7%)	146 (81.1%)	88 (82.2%)
Unfavorable outcome for 1–4	111 (18.3%)	34 (18.9%)	19 (17.8%)
Mortality	74 (12.23%)	13 (7.2%)	10 (9.3%)
Death within one month (n, %)			
Yes	74 (12.2%)	7 (3.9%)	7 (6.5%)
No	531 (87.8%)	173 (96.1%)	100 (93.5%)
CT characteristics at admission			
midline shift (n, %)			
Yes	91 (15.0%)	28 (15.6%)	13 (12.1%)
No	514 (85.0%)	152 (84.4%)	94 (87.9%)
Intracranial lesion (n, %)			
traumatic subarachnoid hemorrhage	353 (58.3%)	109 (60.1%)	55 (51.4%)
epidural hematoma	104 (17.2%)	25 (13.9%)	10 (9.3%)
subdural hematoma	333 (55.0%)	98 (54.4%)	57 (53.3%)
intraparenchymal lesion	304 (50.2%)	93 (51.7%)	55 (51.4%)
Lesion size ≥ 25 mL (n, %)			
yes	81 (13.4%)	19 (10.6%)	12 (11.2%)
no	524 (86.6%)	161 (89.4%)	95 (88.8%)
Basal cistern (n, %)			
normal	462 (76.4%)	141 (78.3%)	84 (78.5%)
compression	86 (14.2%)	23 (12.8%)	13 (12.1%)
occlusion	57 (9.4%)	16 (8.9%)	10 (9.3%)
Marshall classification on admission CT (n, %)			
I–II	397 (65.6%)	115 (63.9%)	74 (69.2%)
III–IV	48 (7.9%)	18 (10.0%)	11 (10.3%)
V–VI	175 (28.9%)	47 (26.1%)	22 (20.6%)
Laboratory examination at admission			
hemoglobin level (G/L) (mean ± sd)	131.7 ± 20.2	133.0 ± 20.0	129.7 ± 20.4
blood glucose level (MMOL/L) (mean ± sd)	7.9 ± 3.2	7.9 ± 3.6	7.9 ± 3.3
white blood cell count (×10^9^/L) (mean ± sd)	11.9 ± 5.3	12.5 ± 5.5	11.8 ± 5.3
monocyte count (×10^9^/L) (mean ± sd)	0.6 ± 0.6	0.6 ± 0.4	0.57 ± 0.43
monocyte ratio (×100%) (mean ± sd)	5.1 ± 2.3	4.9 ± 2.5	5.2 ± 3.2
neutrophil count (×10^9^/L) (mean ± sd)	10.0 ± 5.2	10.5 ± 5.4	10.2 ± 6.0
lymphocyte count (×10^9^/L) (mean ± sd)	1.4 ± 1.2	1.3 ± 1.0	1.4 ± 1.8
lactate level (MMOL/L) (mean ± sd)	2.2 ± 1.5	2.2 ± 1.3	2.1 ± 1.3

**Table 2 jcm-11-00974-t002:** Multivariable logistic regression analysis of the ability of the selected indicators.

Intercept and Variable	Prediction Ability		
	*β*	Odds Ratio (95% CI)	*p*-Value
Age	1.563	4.772 (2.019–11.281)	<0.001
GCS score of 3–8 points	—	—	0.013
GCS score of 9–12 points	−0.711	0.491 (0.148–1.672)	0.245
GCS score of 13–15 points	−1.754	0.173 (0.052-0.580)	0.004
Normal pupil	—	—	0.001
Unilateral pupil reaction	2.398	11.004 (1.089–111.151)	0.042
No pupil reaction	2.685	14.663 (3.131–68.660)	0.001
Hypotension (≤90 mmHg)	1.445	4.240 (1.250–14.380)	0.020
Midline shift (≥5 mm)	1.607	4.986 (1.693–14.688)	0.004
Intracerebral hematoma	0.497	1.645 (0.677–3.995)	0.272
Subarachnoid Hematoma	1.352	3.864 (1.053–14.186)	0.042
Basal cistern—Normal	—	—	0.063
Basal cistern—Compression	1.227	3.411 (1.205–9.655)	0.021
Basal cistern—Occlusion	1.216	3.373 (0.755–15.062)	0.111
Glucose level (>8.1 mmol/L)	0.448	1.565 (0.674–3.636)	0.298
Monocyte count (>0.59 × 109/L)	1.381	3.977 (1.640–9.643)	0.002

## Data Availability

The datasets generated and/or analyzed during the current study are not publicly available due to the inclusion of personal information but are available from the corresponding author upon reasonable request.

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
