# Peer review of "Chinese Admission Warning Strategy for Predicting the Hospital Discharge Outcome in Patients with Traumatic Brain Injury"

_jcm, 2022, doi:10.3390/jcm11040974_

Round 1

Reviewer 1 Report

This manuscript is an interesting content that provides a very good tool for evaluating patients prognosis and predicting it initially.

Predicting the prognosis of TBI patients is an important attempt as planning an initial treatment strategy.

Please response the followings:

  1. Please add contents about references of comparison with basic multiple trauma score such as RTS, AIS (ISS), TRISS etc
  2. in the case of elderly patients mentioned in the discussion text briefly, another factors (medical history, medications) other than mentioned predictors has a great influence on prognosis.  what is authors' comment?
  3.  there are several references to predictors of TBI prognosis that include regional characteristics (Asia, American, African, European..)  Please mention the author's thoughts on this.
  4. please correct  the plural "from strategys to strategies" in conclusion.

Thank you for your submission.  sincerely 

Author Response

This manuscript is an interesting content that provides a very good tool for evaluating patient’s prognosis and predicting it initially.

Predicting the prognosis of TBI patients is an important attempt as planning an initial treatment strategy.

Response:

Thanks for your comments and affirmation.

We have revised the manuscript according to your comments.

The revisions were marked yellow in manuscript.

Please response the followings:

  1. Please add contents about references of comparison with basic multiple trauma score such as RTS, AIS (ISS), TRISS etc

Response:

Thanks for your valuable comments.

We had added the contents about the comparison of our strategy with basic multiple trauma scores:“Although several basic multiple trauma scores such as revised trauma score (RTS), injury severity score (ISS), exponential injury severity score (EISS), and traumatic injury mortality prediction (TRIMP) have been widely accepted, they are so influenced by the anatomical index and the accuracy of specific anatomical injury site (brain injury) is limited29.”

The revisions were marked yellow in manuscript, “Discussion section”.

  1. in the case of elderly patients mentioned in the discussion text briefly, another factors (medical history, medications) other than mentioned predictors has a great influence on prognosis.  what is authors' comment?

Response:

Thanks for your valuable comments.

As is shown in “Feature selection and admission warning strategy developing” and “Supplementary materials”, Another factors (medical history, medications) are not nonzero coefficients in LASSO logistic regression analysis. Therefore, these indicators are not included in the construction of strategy.

These mentioned were marked yellow in manuscript.

  1. there are several references to predictors of TBI prognosis that include regional characteristics (Asia, American, African, European.)  Please mention the author's thoughts on this.

Response:

Thanks for your valuable comments.

As is shown in Introduction section:“In China, from the initial admission to hospitalization in a specialized department, there are often required procedures for emergency triage. During this period, neurologists are often pressed to prognosticate patient outcomes such as mortality or unfavorable outcome during an emergency consultation.” Such a process may unique to China, and the initial intention of this research is to achieve resource optimization and non-invasive prediction.

This study was suitable for Chinese national conditions, which has specific clinical application value in China

  1. please correct the plural "from strategys to strategies" in conclusion.

Response:

Thanks for your valuable comments.

The revisions were marked yellow in manuscript.

Reviewer 2 Report

The paper describes a multiparametric predictive model for outcomes in TBI based on clinical baseline findings, CT imaging and laboratory tests. All input variables come from established tests or examinations and are commonplace in A&E. The results obtained with the model are very promising giving C-index in the range of 0.93 to 0.98. The developed model from 3 Chines TBI clinics was tested in two further cohorts, data from these cohorts aligned with the original patient cohort.

Points of note

1 Page 5. CT imaging. The authors should specify type of CT, i.e. non-contrast CT vs contrast-CT?

2 Page 11, line 2, LASSP?, should be LASSO.

3 Figure 3. Font used in each panel is too small to be readable, please adjust.

Author Response

The paper describes a multiparametric predictive model for outcomes in TBI based on clinical baseline findings, CT imaging and laboratory tests. All input variables come from established tests or examinations and are commonplace in A&E. The results obtained with the model are very promising giving C-index in the range of 0.93 to 0.98. The developed model from 3 Chines TBI clinics was tested in two further cohorts, data from these cohorts aligned with the original patient cohort.

Response:

Thanks for your comments and affirmation.

We have revised the manuscript according to your comments.

The revisions were marked yellow in manuscript.

Points of note

1 Page 5. CT imaging. The authors should specify type of CT, i.e. non-contrast CT vs contrast-CT?

Response:

Thanks for your valuable comments.

The specify type of CT was non-contrast CT.

The revisions were marked yellow in manuscript.

2 Page 11, line 2, LASSP?, should be LASSO.

Response:

Thanks for your valuable comments.

The revisions were marked yellow in manuscript.

3 Figure 3. Font used in each panel is too small to be readable, please adjust.

Response:

Thanks for your valuable comments.

It is impossible to clearly display all the information because there was too much information in the previous Figure 3. In this revision version, we split the previous Figure into two parts (Figure. 3. Internal validation of the warning strategy, and Figure. 4. External validation of the warning strategy). We also increase the clarity of annotation information.

The revisions were marked yellow in manuscript.

Reviewer 3 Report

The authors of the study addressed an important issue, namely the development and validation of an admission warning strategy for individuals after traumatic brain injury, using data from a multicenter study in China. The manuscript is generally well written, the rationale is clear, and the methods used are appropriate. However, there are some information gaps that should be filled to provide more detailed insight into the methodology.

Introduction:

The introduction is well written and provides the reader with all the necessary background information. The objectives of the study are clearly formulated.

Methods: The methods need some more clarification.

p. 6 (Methods): “All potential predictors were applied to develop our strategy by using the cohort26.”

Please, describe the initial model more carefully:

  • What was the dependent variable?
  • What were the independent variables (including variable transformation, e.g., categorizing the GCS)?

Currently, it is unclear how did you define the initial model (see also my comment to the results).

pp. 5-6 (Statistical analyses): Please, specify the cut-offs for evaluation of the goodness of the model fit. For example, which AUC ranges are acceptable?

p. 6 (Methods): You mention the pairwise comparisons. Did you apply an alpha-level adjustment? If yes, report the adjusted alpha level. If no, please, explain why not.

p. 6 (Methods, subsequently Results): You should report optimism-corrected goodness of fit values for the validation analyses results. If you already do it, please, state it in the methods.

Results:

p. 6 (Participants): Please, briefly describe the core characteristics of the study sample. Did the core sample and the subcohorts (see also the next comment) differ significantly according to the core characteristics (sociodemographic and injury-related factors)?

p. 7 (Table 1): You report the results on subcohorts. However, you don’t mention in the methods, how they were obtained and what for. Please, add this information to the methods.

p. 9 (Results): “Based on the admission baseline characteristics and routine, examination indicators of 605 patients with TBI in the primary cohort were collected 6, 7, 15, 22, 29.”

This sentence belongs to the methods.

p. 11 (Results): What was the goodness of fit of the logistic regression? Does it have any impact on the further analyses?

Discussion:

p. 16 (Discussion): Try to avoid the repetition of the results and consider skipping the links to the Tables and Figures in the discussion

Minor issues:

p. 2 (Abstract)/p.5 (Methods)/ Table 1:

Using of ~ is unusual in reporting ranges, suggest changing it into – (e.g., 5–10).

p. 5 (Methods):

I would add the GOSE rating into the determination of the lower and upper categories and delete “where 1 = died…” to avoid repetition, for example, lower/upper good recovery (7/8)

p. 5 (Methods):

Cite the software, add the R version and R packages used

p. 6 (Methods):

“P-values” should be replaced by p-values (no capitalization)

Do you mean in this sentence that the alpha level was set at 5% or that you report only significant results? Please, clarify.

p. 7 (Results/Table 1/2): Use ≥ instead >=; the sample applies for <=

Author Response

The authors of the study addressed an important issue, namely the development and validation of an admission warning strategy for individuals after traumatic brain injury, using data from a multicenter study in China. The manuscript is generally well written, the rationale is clear, and the methods used are appropriate. However, there are some information gaps that should be filled to provide more detailed insight into the methodology.

Response:

Thanks for your comments and affirmation.

We have revised the manuscript according to your comments.

The revisions were marked yellow in manuscript.

Introduction:

The introduction is well written and provides the reader with all the necessary background information. The objectives of the study are clearly formulated.

Methods: The methods need some more clarification.

  1. 6 (Methods): “All potential predictors were applied to develop our strategy by using the cohort26.”

Please, describe the initial model more carefully:

Response:

Thanks for your valuable comments.

We have changed the way of expression the revised version was “All routine indicators available in the China emergency department were collected during the strategy developing phase”

The revisions were marked yellow in manuscript.

What was the dependent variable? What were the independent variables (including variable transformation, e.g., categorizing the GCS)? Currently, it is unclear how did you define the initial model (see also my comment to the results).

Response:

Thanks for your valuable comments.

I'm sorry that the explanation of our previous manuscript is not clear.

The purpose of this article is to screen the predictors that can be combined through LASSO model. Therefore, the independent variables were the basic clinical characteristics that available in the emergency department, and the variables were the hospital discharge outcome in patients with TBI.

  1. 5-6 (Statistical analyses): Please, specify the cut-offs for evaluation of the goodness of the model fit. For example, which AUC ranges are acceptable?

Response:

Thanks for your valuable comments.

I'm sorry that the explanation of our previous manuscript is not clear.

The purpose of this article was to realize the optimal prediction of TBI prognosis. The independent predictors presented in Figure 3 and Figure 4 were acceptable. On this basis, we provide a paradigm for establishing multimodal prediction model. We used C-index, DCA and AUC to provides intuitive comparison between the optimization of multi-modal than traditional independent predictors.

For C-index and AUC, the high accuracy was defined as C-index > 0.9.

For DCA, the accuracy index of our strategy was high than all LASSO selected independent predictors.

In this revision version, we split the previous Figure into two parts (Figure. 3. Internal validation of the warning strategy, and Figure. 4. External validation of the warning strategy).

The revisions were marked yellow in manuscript.

  1. 6 (Methods): You mention the pairwise comparisons. Did you apply an alpha-level adjustment? If yes, report the adjusted alpha level. If no, please, explain why not.

Response:

Thanks for your valuable comments.

I'm sorry that the explanation of our previous manuscript is not clear.

The raw data of the Pairwise comparison of ROC curves is description as shown below:

Model ~ Age

Difference between areas 

0.348

Standard Error a

0.0291

95% Confidence Interval

0.291 to 0.405

z statistic

11.991

Significance level

P < 0.0001

Model ~ GCS

Difference between areas 

0.0592

Standard Error a

0.0143

95% Confidence Interval

0.0312 to 0.0872

z statistic

4.140

Significance level

P < 0.0001

Model ~ MONO

Difference between areas 

0.294

Standard Error a

0.0275

95% Confidence Interval

0.240 to 0.348

z statistic

10.681

Significance level

P < 0.0001

Model ~ GLU

Difference between areas 

0.196

Standard Error a

0.0211

95% Confidence Interval

0.154 to 0.237

z statistic

9.280

Significance level

P < 0.0001

Model ~ Marshall

Difference between areas 

0.127

Standard Error a

0.0171

95% Confidence Interval

0.0935 to 0.161

z statistic

7.423

Significance level

P < 0.0001

Age ~ GCS

Difference between areas 

0.289

Standard Error a

0.0343

95% Confidence Interval

0.222 to 0.356

z statistic

8.428

Significance level

P < 0.0001

Age ~ MONO

Difference between areas 

0.0548

Standard Error a

0.0470

95% Confidence Interval

-0.0373 to 0.147

z statistic

1.167

Significance level

P = 0.2434

Age ~ GLU

Difference between areas 

0.153

Standard Error a

0.0371

95% Confidence Interval

0.0800 to 0.226

z statistic

4.115

Significance level

P < 0.0001

Age ~ Marshall

Difference between areas 

0.221

Standard Error a

0.0346

95% Confidence Interval

0.154 to 0.289

z statistic

6.396

Significance level

P < 0.0001

GCS ~ MONO

Difference between areas 

0.234

Standard Error a

0.0301

95% Confidence Interval

0.175 to 0.294

z statistic

7.779

Significance level

P < 0.0001

GCS ~ GLU

Difference between areas 

0.136

Standard Error a

0.0230

95% Confidence Interval

0.0913 to 0.182

z statistic

5.920

Significance level

P < 0.0001

GCS ~ Marshall

Difference between areas 

0.0679

Standard Error a

0.0236

95% Confidence Interval

0.0217 to 0.114

z statistic

2.881

Significance level

P = 0.0040

MONO ~ GLU

Difference between areas 

0.0980

Standard Error a

0.0319

95% Confidence Interval

0.0355 to 0.160

z statistic

3.072

Significance level

P = 0.0021

MONO ~ Marshall

Difference between areas 

0.167

Standard Error a

0.0315

95% Confidence Interval

0.105 to 0.228

z statistic

5.280

Significance level

P < 0.0001

GLU ~ Marshall

Difference between areas 

0.0686

Standard Error a

0.0284

95% Confidence Interval

0.0129 to 0.124

z statistic

2.415

Significance level

P = 0.0157

  1. 6 (Methods, subsequently Results): You should report optimism-corrected goodness of fit values for the validation analyses results. If you already do it, please, state it in the methods.

Response:

Thanks for your valuable comments.

I'm sorry that the explanation of our previous manuscript is not clear.

To knowledge, the evaluation model is good or bad in two aspects. One is the goodness of fit values. The common evaluation indexes mainly include R square, -2logl, AIC, BIC, etc.; The other is the prediction accuracy of the model. In the prediction model, more attention should be paid to the prediction accuracy because the main purpose of modeling is for prediction. For example, C-index belongs to the prediction accuracy in the model evaluation index. It is worth noting that we conducted the Bootstrapping methods(B=1000 repetitions)before modeling(0.9748729;0.9663948~0.983359), and Calibration curve has been plot that shown in Figure 3 and Figure 4. Therefore, we reported the possibility threshold of 0.01 for the primary cohort, and at thresholds of 0.02-0.83 and 0.01-0.88 for the two subcohort via DCA.

Results:

  1. 6 (Participants): Please, briefly describe the core characteristics of the study sample. Did the core sample and the subcohorts (see also the next comment) differ significantly according to the core characteristics (sociodemographic and injury-related factors)?

Response:

Thanks for your valuable comments.

I'm sorry that the explanation of our previous manuscript is not clear.

We added the descriptions in the results section of this manuscript: In total, 892 patients with TBI were assessed for eligibility between January 2018 and May 2019, of which no significant difference according to the clinical characteristics between the internal validation cohort (605 patients) and the two external validation cohorts (180 and 107 patients).

The revisions were marked yellow in manuscript.

  1. 7 (Table 1): You report the results on subcohorts. However, you don’t mention in the methods, how they were obtained and what for. Please, add this information to the methods.

Response:

Thanks for your valuable comments.

I'm sorry that the explanation of our previous manuscript is not clear.

We had added “These subcenters were randomly divided into three independent cohorts: one training and internal validation cohort, and two external validation cohort. The inclusion criteria for three cohorts were:” …” In the validation phase, we plotted calibration curves to determine the difference between predicted value and actual value, and measured Harrell’s C-index to evaluate the performance of our strategy (developed in primary cohort).” …. “The same process is carried out in three cohort.”

The revisions were marked yellow in manuscript.

  1. 9 (Results): “Based on the admission baseline characteristics and routine, examination indicators of 605 patients with TBI in the primary cohort were collected 6, 7, 15, 22, 29.”

This sentence belongs to the methods.

Response:

Thanks for your valuable comments.

We have deleted this sentence in this section and moved it to the method section.

  1. 11 (Results): What was the goodness of fit of the logistic regression? Does it have any impact on the further analyses?

Response:

Thanks for your valuable comments.

The purpose of this article was to provide a paradigm for establishing multimodal prediction model and realizing the depth analysis of traditional indicators. The independent predictors presented in Figure 3 and Figure 4 were acceptable,as is shown in manuscript,the AUC of our strategy was highest among all predictors. The evaluation model is good or bad in two aspects. One is the goodness of fit values. The other is the prediction accuracy of the model.

In the future, we will make up for the lack of research in this area.

Thanks again for your suggestion.

Discussion:

  1. 16 (Discussion): Try to avoid the repetition of the results and consider skipping the links to the Tables and Figures in the discussion

Response:

Thanks for your valuable comments. We had removed the repetition of the results and the links to the Tables and Figures in the discussion.

Minor issues:

  1. 2 (Abstract)/p.5 (Methods)/ Table 1:

Using of ~ is unusual in reporting ranges, suggest changing it into – (e.g., 5–10).

Response:

Thanks for your valuable comments. We had change all “~” into “-”.

The correction was completed and marked in yellow.

  1. 5 (Methods):

I would add the GOSE rating into the determination of the lower and upper categories and delete “where 1 = died…” to avoid repetition, for example, lower/upper good recovery (7/8)

Response:

Thanks for your valuable comments. The TBI patient outcomes were dichotomized into “favorable outcome” defined as a GOSE score of 5-8 points and “unfavorable outcome” defined as a GOSE score of 1-4 points.

The correction was completed and marked in yellow.

  1. 5 (Methods):

Cite the software, add the R version and R packages used

Response:

Thanks for your valuable comments. Statistical computing and graphic were performed with R software (version 3.5.0; https://www.r-project.org) and SPSS 22.0 with p value less than 0.05 considered as statistical significance.

This correction was added in manuscript and marked in yellow.

  1. 6 (Methods):

“P-values” should be replaced by p-values (no capitalization)

Response:

Thanks for your valuable comments.

The correction was completed and marked in yellow.

Do you mean in this sentence that the alpha level was set at 5% or that you report only significant results? Please, clarify.

Response:

Thanks for your valuable comments, we report only significant results in this study.

This correction has been clarified in manuscript and marked in yellow.

  1. 7 (Results/Table 1/2): Use ≥ instead >=; the sample applies for <=

Response:

Thanks for your valuable comments.

This correction has been clarified in manuscript and marked in yellow.
